# DNA assembly for nanopore data storage readout

Randolph Lopez[1,2], Yuan-Jyue Chen[3], Siena Dumas Ang[3], Sergey Yekhanin[3], Konstantin Makarychev[3], Miklos Z Racz[3], Georg Seelig [2,4,5], Karin Strauss[3] & Luis Ceze [5]

Synthetic DNA is becoming an attractive substrate for digital data storage due to its density, durability, and relevance in biological research. A major challenge in making DNA data storage a reality is that reading DNA back into data using sequencing by synthesis remains a laborious, slow and expensive process. Here, we demonstrate successful decoding of 1.67 megabytes of information stored in short fragments of synthetic DNA using a portable nanopore sequencing platform. We design and validate an assembly strategy for DNA storage that drastically increases the throughput of nanopore sequencing. Importantly, this assembly strategy is generalizable to any application that requires nanopore sequencing of small DNA amplicons.

[1] Department of Bioengineering, University of Washington, Seattle, WA 98105, USA. [2] Molecular Engineering & Sciences Institute, University of Washington, Seattle, WA 98195, USA. [3] Microsoft Research, Redmond, WA 98052, USA. [4] Department of Electrical & Computer Engineering, University of Washington, Seattle, WA 98195, USA. [5] Paul G. Allen School for Computer Science & Engineering, University of Washington, Seattle, WA 98195, USA. Correspondence and requests for materials should be addressed to L.C. (email: luisceze@cs.washington.edu)

The digital revolution has resulted in the exponential growth of electronic data storage. During the past three decades, the amount of digital information has doubled every 2.5 years and it is expected to reach 375 exabytes by 2040[1,2]. Although not all data needs to be stored, significant portions do. In anticipation of this significant increase in data storage demand, synthetic DNA has been widely studied as a promising storage medium for data archival[1,3–11]. DNA offers ultrahigh information density capabilities in the order of hundreds of petabytes per gram[6,8] and under proper conditions it can retain information for millions of years[10,12]. Furthermore, technologies that enable high-throughput reading and writing of synthetic DNA have been rapidly evolving in parallel with the advent of the genomics revolution[13]. Nevertheless, several challenges remain unaddressed before DNA storage is able to meet the existing demand for data storage and become a cost-effective alternative to silicon.

DNA sequencing is used to read the information encoded in DNA back into digital bits. Currently, sequencing by synthesis (SBS), as commercialized by Illumina, is the leading technology for high-throughput sequencing[13]. In previous work, we demonstrated the ability to write over 200 megabytes of information in about 2 billion nucleotides while recovering all data using random access without any bit errors using Illumina SBS technology[7]. Our group and others have demonstrated random access by selective PCR amplification of a given file without having to read all the data stored in a particular DNA pool[3,5,8,9]. Despite its low error rate and high throughput, SBS technology has several shortcomings in the context of DNA storage. In its current form, it requires bulky and expensive instrumentation and access to sequencing data is delayed until completion of the entire sequencing run.

Nanopore sequencing, as commercialized by Oxford Nanopore Technologies (ONT), offers a sequencing alternative that is portable, inexpensive and automation-friendly, resulting in a better option for a real-time read-head of a molecular storage system[14–18]. Specifically, ONT MinION is a four-inch long USB-powered device containing an array of 512 sensors, each connected to four biological nanopores, capable of producing up to 15 gigabases of sequencing output per flowcell. Each nanopore is built into an electrically resistant artificial membrane. During sequencing, a single strand of DNA passes through the pore resulting in a change in the current across the membrane. This electrical signal is processed in real time to determine the sequence identity of the DNA strand. In the context of DNA storage, real-time sequencing enables the ability to sequence until sufficient coverage has been acquired for successful decoding without having to wait for an entire sequencing run to be completed. Moreover, nanopore sequencing offers a clear single-device throughput scalability roadmap via increased pore count, which is very important for viability of DNA data storage.

Nanopore sequencing presents unique challenges to decoding information stored in synthetic DNA. In addition to a significantly higher error rate compared to SBS, nanopore sequencing of short DNA fragments results in lower sequencing throughput due to inefficient pore utilization. Previous work by Yazdi et. al. demonstrated a DNA storage workflow where 17 unique large DNA fragments (~1000 bp.) encoding for a 3kB file were synthesized, and subsequently sequenced and decoded using ONT MinION platform[9]. However, existing scalable approaches for writing synthetic DNA rely on parallel synthesis of millions of short oligonucleotides (i.e., 100–200 bases in length) where each oligonucleotide contains a fraction of an encoded digital file. We find that sequencing of such short fragments results in significantly lower sequencing throughput in the ONT MinION. This limitation hinders the scalability of nanopore sequencing for DNA storage applications.

In this work, we demonstrate an approach for decoding information stored in DNA that combines random-access, DNA assembly and nanopore sequencing. In total, we amplified, concatenated and successfully decoded three files totaling 1.67 megabytes of digital information stored in 111,499 oligonucleotides (Fig. 1a). Our work results in a 2-order of magnitude increase in demonstrated sequencing and decoding capacity using nanopore sequencing for a DNA storage application. Our real-time sequencing implementation for DNA storage provides a faster and more flexible alternative to decoding digital files encoded in DNA where sequencing can be carried out until enough coverage has been acquired for decoding. To achieve this, we implement a strategy that enables both random access and molecular assembly of a given DNA file stored in short oligonucleotides (150 bp) into large DNA fragments containing up to 24 oligonucleotides (~5000 bp). We evaluate Gibson Assembly and Overlap-Extension Polymerase Chain Reaction (OE-PCR) as suitable alternatives to iteratively concatenate and amplify multiple oligonucleotides in order to generate large sequencing reads. To decode the files, we implement a consensus algorithm capable of handling high error rates associated with nanopore sequencing. We demonstrate this approach by amplifying, assembling and sequencing four different files stored in DNA with significant improvements in capacity (bases/flowcell) and overall throughput (bases/second). We decode files with a minimum sequencing coverage as low as 22×, compared to 36× in our previous work[7]. Furthermore, our Gibson Assembly concatenation strategy is generalizable to any short amplicon sequencing application where higher nanopore sequencing throughput is desirable.

## Results

**Consecutive Gibson assembly for DNA storage file retrieval.** The overall yield and quality of a nanopore sequencing run is dependent on the molecular size of the DNA to be sequenced. DNA molecules translocate through the pore at a rate of 450 bases/sec while it can take between 2–4 s for a pore to capture and be occupied by the next DNA molecule. Therefore, short DNA molecules result in a higher number of unoccupied pores over time, which increases the rate of electrolyte utilization above the membrane. This results in a faster loss in polarity and lower sequencing capacity and overall throughput. ONT estimates that the optimal DNA size to maximize sequencing yield is around 8 kilobases while the minimum size is 200 bases. Below 200 bases, event detection and basecalling is not possible. Therefore, a PCR-based random-access strategy for sequencing of the files encoded in 150 bp DNA oligonucleotides would have resulted in very low sequencing yield and limited decoding capabilities.

Thus, we applied Consecutive Gibson Assembly to assemble large sequencing reads from short amplicons for the purpose of enabling higher nanopore sequencing throughput (Fig. 2a). Gibson Assembly is an isothermal and single-reaction method for assembly of multiple DNA sequences first developed by Daniel Gibson et. al. in 2009[19]. Since then, it has become a fundamental tool for the assembly of synthetic gene constructs in molecular cloning and synthetic biology due to its modularity and ease-of-use[20–22]. We re-purposed this tool for DNA storage to enable assembly and random access of digital files encoded in DNA pools containing multiple files in millions of oligonucleotides.

Our decoding process begins with a pool of synthetic DNA containing several files. Each file in the oligo pool consists of a set of 150-bases oligonucleotides with unique 20-nucleotide sequences at their 5′ and 3′ ends for PCR-based random-access retrieval (i.e., file ID) and a 110-nucleotide payload encoding the digital information. To amplify a specific file, PCR reactions are

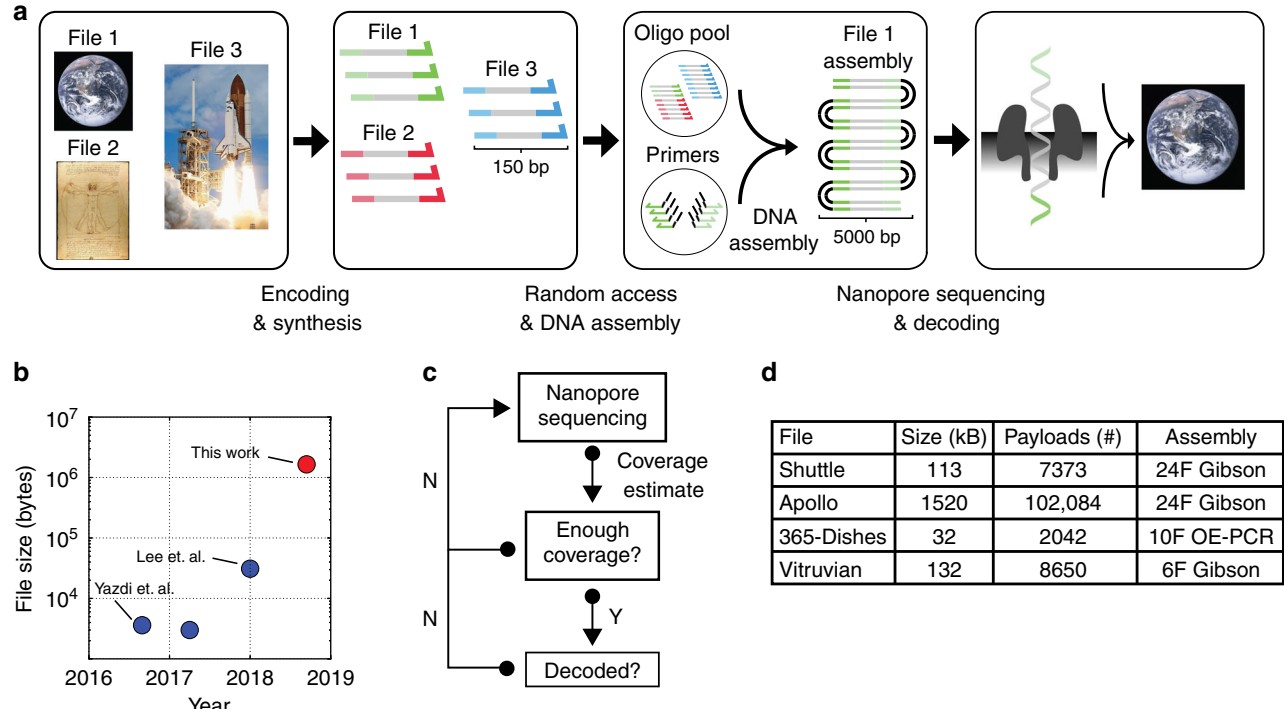

**Fig. 1** Overview of the DNA data storage workflow. **a** The encoding process starts with mapping multiple digital files into 150-nucleotide DNA sequences and sending them for synthesis. Each file has unique sequence addresses at the 5′ and 3′ end of each oligonucleotide for random access retrieval. Using PCR primers containing complementary overhang sequences, a specific file can be amplified and concatenated into long double-stranded DNA molecules suited for ONT Nanopore sequencing. Upon sequencing, a subset of reads with high accuracy are used to decode the selected file. **b** Our assembly and decoding strategy enabled successful decoding of 1.67 MB of digital information stored in DNA using nanopore sequencing. Our work represents a 2-order of magnitude improvement in demonstrated decoding ability using nanopore sequencing for DNA storage. **c** Sequence-until diagram. Nanopore sequencing enables real-time coverage estimation for decoding of digital files store in DNA. This enables the user to generate reads until coverage is enough for successful decoding. Upon decoding, a different file can be sequenced in the same flowcell or the sequencing run can be stopped and resumed later on. **d** Four different files encoded in DNA were amplified, assembled and sequenced using ONT MinION platform. We implemented overlap-extension PCR and Gibson Assembly to build assemblies of 6, 10, or 24 fragments for each file

carried out with primer sets designed to amplify a file's ID, thereby enabling random access retrieval (Fig. 2b). To enable assembly, each file is amplified with multiple PCR reactions where each PCR reaction contains primer sets with the same file ID but with sequentially overlapping DNA overhangs such that the amplified fragments can be combined in a Gibson Assembly in a later step. Therefore, the number of PCR reactions depends on the chosen assembly size for a file.

Overhang sequence design was performed using a nucleic acid thermodynamic simulation software (NUPACK[23]) to avoid primers with self-binding structures and cross-talk between orthogonal overhangs among other constraints (Supplementary Fig. 2). First, we generated a random set of 30-nucleotide DNA sequences with GC content between 40% and 60%, while avoiding 4-nucleotide repeats of G or C. Subsequently, we used NUPACK to estimate intramolecular secondary structure probability and selected those sequences with minimal secondary structure. This selection step was necessary to enable PCR reactions with comparable amplification efficiencies, which facilitated the generation of the assembly fragments. Next, we evaluated our overhang primer set for orthogonality by estimating binding probability across all primer pairs. Based on this cross-talk mapping, we selected a subset of primers that exhibit minimum cross-talk among all pairs. We repeated this process to create overhangs for every new file to be assembled since each file contained different primer sequences (Supplementary Table 1). We expect that selecting an overhang sequence subset with minimal cross-talk can result in higher efficiency in the assembly

process. We used this DNA sequence design method to generate overhang sequences for each file assembly.

After PCR amplification of each file with its corresponding priming pairs, PCR products are combined, purified using magnetic beads and combined into a Gibson Assembly reaction. During the Gibson reaction, the exonuclease creates single-stranded 3′ overhangs that enable the annealing of fragments that share complementarity at one end while the polymerase fills in gaps within each annealed fragment. Finally, a DNA ligase seals nicks in the assembled DNA. The product of the Gibson assembly is then amplified using primers corresponding to unique overhang sequences present at the ends of the assembly product. We implemented this approach to generate a 6-fragment assembly for each file. Using agarose gel electrophoresis, we found that the amplified Gibson Assembly product was the expected size (1110 base pairs) with very small amounts of secondary products (Fig. 2c).

To generate even longer fragments, we performed a second Gibson Assembly iteration. The first assembly product was PCR-amplified using primers containing a second set of complementary overhangs in separate PCR reactions. As with the first assembly, the PCR products were purified, combined into a Gibson Assembly reaction and amplified using primers corresponding to unique sequences present at the ends of the assembly product. We implemented this approach to generate a 4-fragment second assembly resulting in 24-fragments of each individual oligonucleotide in the final product. Using agarose gel electrophoresis, we found that the amplified second assembly product

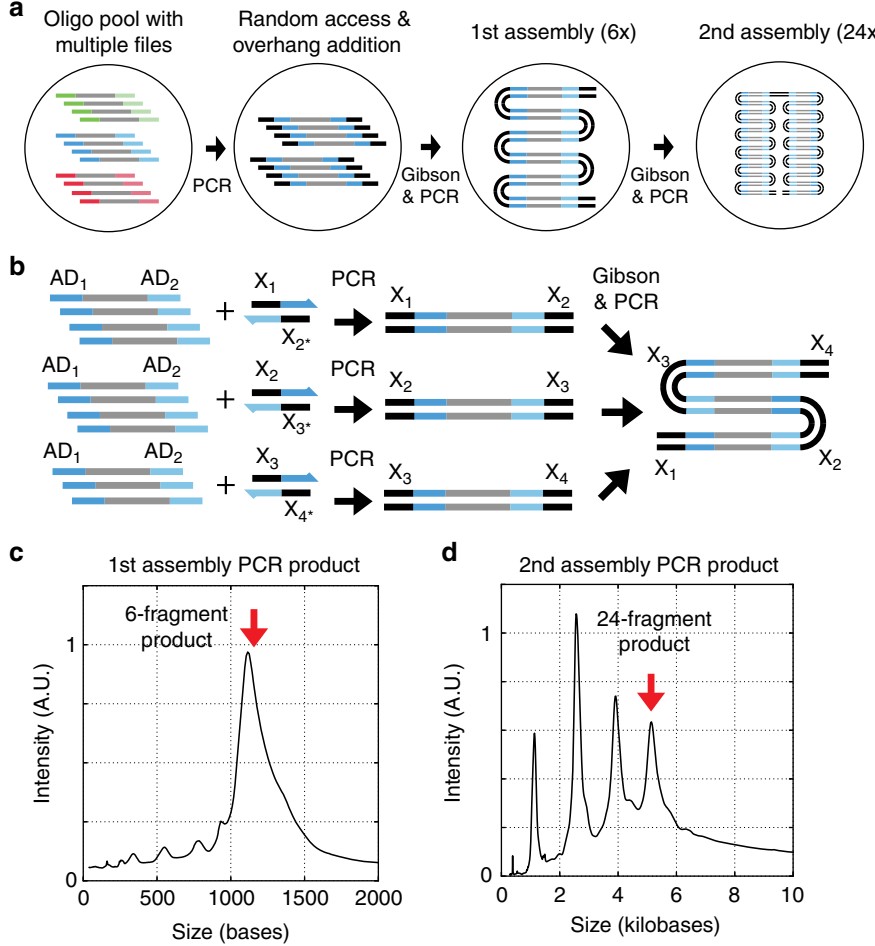

**Fig. 2** Random access and sequential Gibson strategy for DNA storage. **a** Our sequential Gibson Assembly strategy enables random access and concatenation of any particular file from a pool of DNA oligonucleotides. We demonstrated up to 24-fragment assembly by performing two sequential assemblies of 6 and 4 fragments, respectively. **b** First, a given file is PCR-amplified using primers specific to its address (AD1 and AD2) and containing overlapping overhang sequences ($X_n$). For a three-fragment assembly, three separate PCR amplification reactions are carried out, each containing primer pairs with different overhangs based on a sequential assembly design ($X_1$ and $X_{2^*}$, $X_2$ and $X_{3^*}$, and $X_3$ and $X_{4^*}$). Upon amplification, products are purified and combined by Gibson Assembly. The assembly product can consist of any combination of oligonucleotides in a given file pool separated by ordered overhangs with a consistent directionality (e.g., X1-AD1-Payload1-AD2-X2-AD1-Payload2-AD2-X3-AD1-Payload3-AD2-X4). This product is then amplified by using primers specific to ends of the assembly ($X_1$ and $X_{4^*}$) and can be used as the starting material for a second assembly. **c** Gel electrophoresis size distribution corresponding to a PCR amplification of a 6-fragment Gibson Assembly. The expected band size was 1110 bp. **d** Gel electrophoresis size distribution corresponding to a PCR amplification of a 24-fragment second Gibson Assembly. The expected band size was 4590 bp. The correct fragment was gel extracted and used for nanopore sequencing

was the expected size (4590 base pairs). However, we also observed significant quantities of shorter DNA fragments that appear to correspond to smaller assembly sizes (Fig. 2d). The fragment of correct size was gel extracted and used as input material for ONT nanopore sequencing.

We implemented Consecutive Gibson Assembly and nanopore sequencing on three files: a picture of the space shuttle (Shuttle, 115 kilobytes), a picture of the earth viewed from the Apollo 17 mission (Apollo, 1.5 megabytes) and a picture of Leonardo da Vinci's Vitruvian Man (Vitruvian, 132 kilobytes). The Vitruvian Man had an encoding allowing for homopolymers in the DNA sequence while the other two files did not. The Space-Shuttle file and the Apollo file were concatenated into 24-fragment assemblies while the homopolymer file was concatenated into a 6-fragment assembly. In a previous publication from our group, we had demonstrated successful decoding of all these files without assembly using an Illumina sequencing platform[7].

Consecutive Gibson Assembly enables concatenation of DNA amplicons with identical primer regions into larger DNA fragments. The assembly size is determined by the number of fragments with unique complementary overhangs at each end. Furthermore, since the assembly product contains unique sequences at its end, a final amplification step can target the intended assembly. Furthermore, we demonstrated that this process can be iterative: the product of the first assembly can then become the initial fragment for a second assembly. Even though the sample preparation is significant, we found the approach to be flexible and robust for assembling oligonucleotides from different files.

**Payload concatenation using OE-PCR.** Although Consecutive Gibson Assembly can concatenate multiple oligonucleotides, we found it difficult to assemble more than six fragments in a single reaction. We observed that the relative proportion of side products increased with the number of attempted fragments in the assembly. Since all fragments in the assembly contain a conserved file ID region, it is possible that base-pairing across this region

resulted in the generation of these side products. Furthermore, the iterative Gibson Assembly strategy requires significant sample preparation, which hinders its applicability in an end-to-end DNA storage workflow.

As an alternative to Gibson Assembly, we explored the use of Overlap-Extension PCR (OE-PCR) to assemble multiple oligonucleotides into larger DNA fragments (Fig. 3a and Supplementary Table 2). In the context of molecular cloning, OE-PCR is used as a simple approach to insert DNA fragments into plasmids or to join different gene fragments together[24,25]. To repurpose this approach for DNA storage, our strategy first splits the fixed-size DNA fragments of a file into multiple groups where each group has a unique front ID and a unique back ID. In the first step of PCR, overlapping sequences between each DNA group can be created by using primers containing a 5′ overhang complementary to the molecule it is joined to (Fig. 3b). All amplified DNA groups are mixed together, and DNA groups with overlapping regions can be fused together via PCR with N cycles (N equals to the number of DNA groups). Finally, the outermost primers are used to selectively amplify the full length of multiple-fused DNA.

We selected a text file corresponding to 365 Foreign Dishes (Dishes, 32 kB), a vintage book originally published in 1908. We encoded this file into 2042 oligonucleotides and split into 10 groups (i.e., each group has about 206 oligonucleotides, and each group has its own unique IDs). We successfully used OE-PCR to assemble 10 fragments into a long DNA fragment with no visible side products (Fig. 3c). The absence of side products reduced the sample preparation process compared to the Consecutive Gibson Assembly approach since no gel-purification was required. The assembly product was purified using magnetic beads and then used as input material for ONT nanopore sequencing.

**Nanopore sequencing and decoding.** Each file was amplified, assembled and then sequenced in separate nanopore flowcells (R9.4 chemistry). Each sequencing run generated reads for 48 h until completion. The Earth-Apollo file was sequenced using two MinION flowcells to generate enough reads for successful

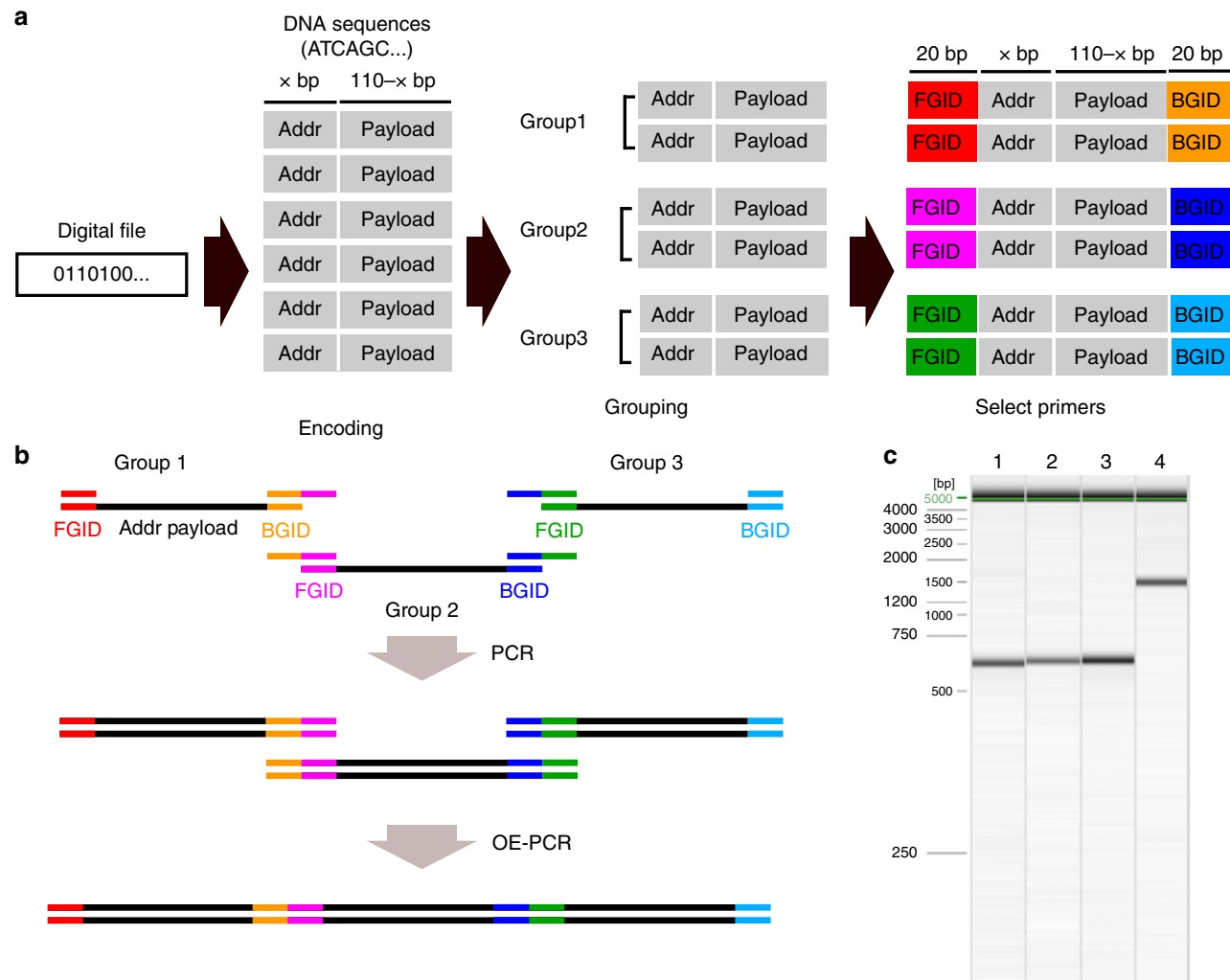

**Fig. 3** Random access and OE-PCR strategy for DNA storage **a** First, a digital file is encoded using Reed–Solomon code to increase robustness to errors. The binary data then get broken into fixed-size payloads and mapped to 110-bp DNA nucleotides with addressing information (Addr). These DNA fragments are split into multiple groups, each group is given unique primers as group ID (FGID front group ID, BGID back group ID). **b** To retrieve a particular file, each group is amplified with primers containing overhangs overlapping with each adjacent group primers. Subsequently, all the groups can be combined and amplified using primers corresponding to the end (e.g., FGID in the group 1 and BGID in the group 3) to form the assembly product using overlap-extension PCR. **c** Bioanalyzer traces corresponding to the assembly of group 1–2–3 (lane 1), group 4–5–6 (lane 2), group 7–8–9 (lane 3), and group 1 through 10 (lane 4). We found no observable by-products in the assembly of up to 10 groups demonstrating the scalability of this approach

decoding. Sample preparation was done accordingly to ONT instructions for sequencing using 1D$^2$ chemistry and only 1D$^2$ reads were used for decoding. We analyzed these reads to evaluate the throughput, quality and decoding potential of each run.

For the 1.5 MB Earth-Apollo file, two sequencing runs generated a total of 267,152 1D$^2$ reads for analysis and decoding. The size distribution of nanopore reads corresponded closely to the gel-extracted input DNA, indicating that most reads corresponded to the full assembly (Fig. 4a). We aligned each read to the expected payloads and we found that on average each read contributed to 18 alignments, close to the ideal maximum of 24 (Fig. 4b). This resulted in a 43X average sequencing coverage for 102,084 reference payloads (Fig. 4c). For comparison, 4.4 M reads would have been necessary to achieve the same sequencing coverage without any assembly. We found a wide distribution in the quality score of all sequencing reads by analyzing the Phred quality score estimated by ONT (Fig. 4d). The average Phred quality score per read was 15.33, corresponding to an estimated error rate probability of 2.93%. Additionally, we estimated the overall error rate based on our alignment and we found an error rate of 6.87%. For insertions and deletions, we found no significant bias across bases. However, substitution error rates were significantly higher between purines (A to G and G to A) and pyrimidines (C to T and T to C) (Fig. 4e). We found a similar error distribution across all files sequenced.

We performed an equivalent sequencing analysis for the other three files (Supplementary Figs. 3, 4, and 5) and then we attempted decoding each file using the 1D$^2$ reads as inputs to a decoder algorithm. Since each read contains multiple payload sequences, we performed multiple sequential alignments of the front and back file ID in each read and then selected the sequence in between, which should correspond to the payload sequences. The decoding process starts by clustering these payload sequences based on similarity and finding a consensus between the sequences in each cluster to reconstruct the original sequences. Once a consensus has been reached, these sequences are decoded back to digital data to retrieve the original file. We used a decoder algorithm from our previous work but modified the consensus sequence algorithm to better utilize nanopore sequencing data.

In our previous decoding algorithm[7,26], the consensus sequence is recovered by a process where pointers for payload sequences are maintained and moved from left to right, and at every stage of the process the next symbol of the sequence is estimated via a plurality vote. For payload sequences that agree with plurality, the pointer is moved to the right by 1. But for the sequences that do not agree with plurality, the algorithm classifies whether the reason for the disagreement is a single deletion, an insertion, or a substitution. This is done by looking at the context around the symbol under consideration. Once this is estimated, the pointers are then moved to the right accordingly.

The key difference between our new algorithm and our previous implementation is that in cases when disagreements cannot be classified, we do not drop respective payload sequences from further consideration. Instead, we label such payload sequences as being out of sync and attempt to bring them back at later stages. Specifically, every sequence that is out of sync is ignored for several next steps of the algorithm. However, after those steps, we perform search for a match between the last few bases of the partially constructed consensus sequence and appropriately located short substrings of the payload sequence.

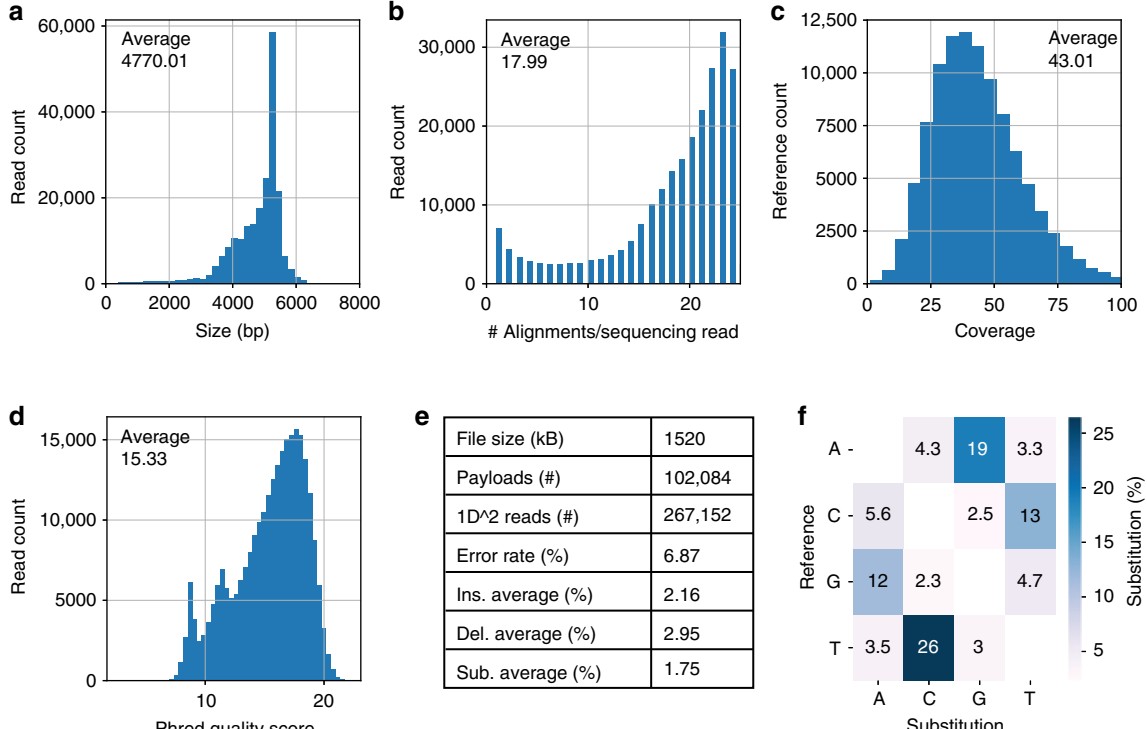

**Fig. 4** Nanopore sequencing analysis for 1.5 MB Apollo file. Two MinION flowcells generated 267,152 1D$^2$ reads of the 24-fragment Gibson Assembly of the Apollo file. **a** Base pair size of sequencing reads matches closely with the assembly size of 4590 bp. **b** We aligned each reference payload sequence to the sequencing reads. Each sequencing read resulting in an average of 17.99 alignments to different payloads. Ideally, each read should have 24 alignments. **c** We found an average sequencing coverage of 43x. Without any assembly, the same number of reads would have resulted in an average sequencing coverage of 2.6x. **d** We estimated raw sequencing quality by aggregating the average Phred quality score in each read. **e** Based on the reads that aligned to payloads, we calculated the average percent error for each base for insertions, deletion, and substitutions. **f** Substitution comparison across different bases revealed strong bias in between purines and pyrimidines

If we discover a match, we move the payload sequence pointer to the corresponding location and drop the out of sync label. This allows the payload sequence pointer to circumvent small groups of adjacent incorrect bases, a feature that was not present in the earlier algorithm. This modification to the consensus algorithm allows us to successfully decode from notably lower coverages because more information from the sequencing reads is used in the process.

We successfully decoded both non-homopolymer files (Apollo and SpaceShuttle files) concatenated using a 24-fragment sequential Gibson Assembly. We were also successful decoding the 365 Dishes file assembled using the 10-fragment OE-PCR strategy. By subsampling the number of reads used for decoding, we found that minimum coverages of 23×, 22×, and 27× were sufficient to decode the Apollo, Shuttle and 365 Dishes files, respectively. Nevertheless, despite having a coverage of 166×, we were unable to decode the Vitruvian Man file, which contained homopolymers within its payload. Even though the overall error rate was similar to that of the non-homopolymer files, we found that nanopore sequencing tends to underestimate the length of homopolymer runs, leading to correlated deletion errors across payload sequences that cannot be corrected by the consensus algorithm. 26% of all original sequences corresponding to the Vitruvian Man file contained a homopolymer run of length 5 or more. Approximately 44% of reads of these runs contained a deletion error, despite the overall deletion rate of just 3%.

We found that increasing the input DNA size resulted in significant improvements in the sequencing throughput. To evaluate this improvement, we quantify sequencing throughput across five sequencing runs of equivalent quality. When comparing sequencing runs based on the input DNA fragment size, we found a modest reduction in the number of reads as the fragment size increased (Fig. 5a). However, we found a 7-fold increase in sequencing throughput across the fragment size range we evaluated (between 600 and 4700 base pairs) (Fig. 5b). This increase in sequencing yield was necessary to enable the successful decoding of a 1.5 MB file while sequencing using only two MinION flowcells.

## Discussion

We demonstrated megabyte-scale decoding of digital files encoded in synthetic DNA using a portable nanopore sequencer. In combination with our robust encoding scheme, our assembly framework enabled sequencing and decoding of a 1.5-megabyte file using only two ONT MinION flowcells. We also demonstrated that this method is compatible with a PCR-based random-access strategy for DNA storage. Our assembly strategy can significantly increase the effective coverage of nanopore sequencing in other applications that require sequencing of short DNA amplicons.

We found that assembling large DNA fragments from short DNA oligonucleotides was crucial for increasing nanopore sequencing throughput and decoding large digital files stored in DNA. However, both assembly methods required additional sample preparation steps compared to our previous work[7] where random access and sample preparation for SBS was enabled by a single PCR reaction from the original pool. As an alternative, we experimented with rolling circle amplification (RCA) since it had been previously described as an easier alternative for building assemblies for nanopore sequencing[27]. However, we found a reduction in sequencing throughput when using the product of an RCA reaction compared to the other assembly methods. A potential explanation for reduction in throughput is that the branched nature of the RCA product hinders the translocation of the DNA through the nanopore resulting in fewer sequencing reads. In addition to these alternatives, there are many other DNA assembly methods that could be adapted into a nanopore sequencing sample preparation workflow and may enable higher sequencing throughput with more efficient sample preparation.

The future of DNA data storage will depend on technologies that enable cost-efficient and fast reading and writing synthetic DNA. While sequencing by synthesis remains widely used for its low error rate and reliability, nanopore sequencing is being rapidly adopted in applications that require long DNA sequencing reads[28], high degree of portability[14], or real-time sequencing[15]. High error rates and limited sequencing throughput remain significant limitations for this technology to be applicable in a scalable DNA storage workflow. Nevertheless, the commercial implementation of this technology is still in its infancy and future technological improvements can enable a nanopore-based read-head for DNA storage to become cost-effective and widely adopted. Specifically, recent advances in basecalling algorithms[29], improved biological nanopores[30], and the emergence of solid-state nanopores[31] promise to further improve the cost and scale of nanopore sequencing.

## Methods

**Encoding files into DNA**. Our encoding process starts by randomizing data to reduce chances of secondary structures, primer–payload non-specific binding, and improved properties during decoding. It then breaks the data into fixed-size payloads, adds addressing information, and applies outer coding, which adds redundant sequences using a Reed–Solomon code to increase robustness to missing sequences and errors. The level of redundancy is determined by expected errors in sequencing and synthesis, as well as DNA degradation. Next, it applies inner

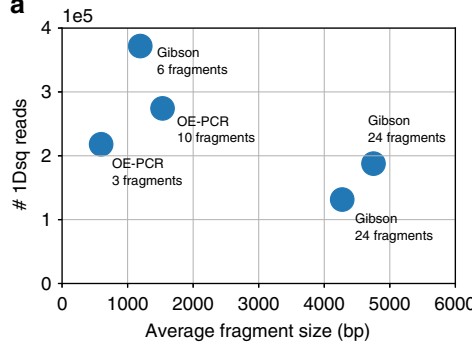
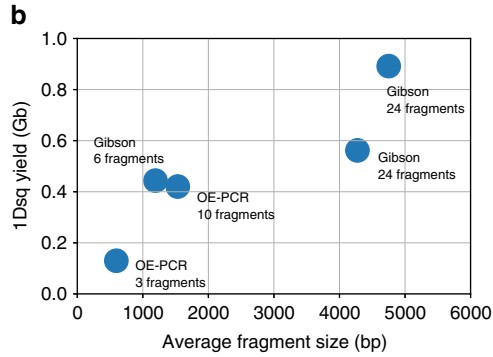

**Fig. 5** Comparison between nanopore sequencing runs with different DNA fragment sizes. We compared five nanopore sequencing runs of equivalent sequencing quality to understand how input DNA size affects sequencing throughput. **a** Number of 1D² sequencing reads generated in different sequencing runs. We found a modest decrease in the number of 1D² reads as the input DNA size increased. **b** 1D² sequencing yield (# of bases) generated in different sequencing runs. We found that large assembly sizes resulted in higher sequencing yield and higher coverage

coding, which converts the bits to DNA sequences. The resulting set of sequences is surrounded by a primer pair chosen from the library based on low level of overlap with payloads. A more detailed description of the encoding algorithm can be found in our previous publication[7]. The resulting collection of sequences is synthesized by Twist Bioscience. Upon arrival, DNA is rehydrated the pool in $1 \times$ TE buffer and stored at $-20\,^{\circ}C$.

**Assembly sequence design**. Overhang sequence design was performed using a nucleic acid thermodynamic simulation software NUPACK v3.0.5. To evaluate primer self-binding, secondary structure analysis was performed using 'pairs' function from NUPACK at $65\,^{\circ}C$ to simulate the temperature of the annealing step in the PCR reaction. Resulting secondary structure predictions are ranked by probability of self-pairing and primers with low secondary structure are selected. To evaluate potential cross-talk between different overhangs in the assembly step, we evaluated cross-talk among all pairs of overhang primers using 'pairs' function from NUPACK at $25\,^{\circ}C$, at 10 nM of each overhang primer. Even though the Gibson Assembly reaction is carried out at $50\,^{\circ}C$, we set the temperature parameter at $25\,^{\circ}C$ for a more stringent simulation. Next, pairs with an equilibrium binding of 1 nM or above were sequentially eliminated until finding a subset of primers with minimal cross-talk.

**PCR amplification**. PCR-based random access and overhang addition of a given file from a DNA pool was performed as following: mix 10 ng of ssDNA pool, 2 μM of forward and reverse primer, 25 μL of 2x Kapa HiFi enzyme mix, 1.25 μL of EvaGreen dye 20X and 20 μL of molecular grade water. After vortexing and centrifuging, the PCR reaction was placed in a Thermo Fisher Scientific Stratagene Mx3005P quantitative PCR instrument with the following protocol: (1) $95\,^{\circ}C$ for 3 min, (2) $98\,^{\circ}C$ for 20 s, (3) $65\,^{\circ}C$ for 20 s, (4) $72\,^{\circ}C$ for 1 min, and (5) go to step 2 a varying number of times until fragment is amplified. All PCR reactions were carried out on a quantitative PCR instrument where amplification was stopped when the reaction reached a plateau phase. All primers were ordered from Integrated DNA Technologies. Gibson Assembly products were amplified using an equivalent protocol, but the extension time was adjusted by adding 1 additional minute for every additional kilobase in size. Gel images corresponding to the Gibson Assembly products after amplification were processed using ImageJ software. Raw gel images corresponding to the first and second assembly are included in the GitHub repository.

**Gibson assembly**. PCR reactions were cleaned-up using KAPA Pure beads (Roche Cat # KK8000) before added to the first Gibson Assembly. PCR reactions were gel-purified using a 1% agarose gel and NEB Monarch DNA Gel Extraction kit for the second Gibson Assembly reactions in order to select out any side products from the first assembly. Gibson Assembly reactions were carried out using NEB Gibson Assembly Master Mix where the fragments were combined at 200 ng of DNA and incubated at $50\,^{\circ}C$ for 1 h. After the assembly, reactions were purified using KAPA Pure beads.

**OE-PCR**. First, each individual fragment was amplified with the following PCR protocol. In a 20 uL reaction, 1 uL of 10 nM single-stranded DNA pool was mixed with 1 uL of 10 uM of the forward primer and 1 uL of 10 uM of the reverse primer, 10 uL of 2X KAPA HIFI enzyme mix, 1 uL of 20X EVA Green, and 6 uL of molecular grade water. The reaction followed a thermal protocol: (1) $95\,^{\circ}C$ for 3 min, (2) $98\,^{\circ}C$ for 20 s, (3) $62\,^{\circ}C$ for 20 s, and (4) $72\,^{\circ}C$ for 15 s. The PCR reaction was performed on a quantitative PCR (qPCR) instrument and stopped before the reaction reached a plateau phase. The length of amplified product was confirmed using a Qiaxcel fragment analyzer, and the sample concentration was measured by Qubit 3.0 fluorometer.

Next, all amplified fragments were mixed with equal molar ratio (1.5 ng for each fragment) together with 10 uL of 2X KAPA HIFI enzyme mix in a total of 20 uL reaction. The reaction followed a standard PCR thermal protocol for N cycles (N is the total number of fragments). After that, 1 uL of the amplified product was mixed with 1 uL of 10 uM of the forward primer, 1 uL of 10 uM of the reverse primer, 10 uL of 2X KAPA HIFI enzyme mix, and 7 uL of molecular grade water. The mixture followed the same thermal protocol as above and stopped when reaching a plateau phase. The final product was verified using a Qiaxcel fragment analyzer to determine the correct size and purity of the assembly.

**ONT MinION sequencing**. Final assembly products were amplified in multiple PCR reactions to generate 1 μg of DNA after gel purification. Purification was carried out with a NEB Monarch DNA Gel Extraction kit, followed by column clean-up with Sigma–Aldrich Sephadex G-25 to remove excess salt. Finally, KAPA Pure beads were used to concentrate the final DNA library to 45 μL for nanopore sequencing. DNA purity and concentration were verified using Thermo Fisher Qubit and NanoDrop instruments. Sequencing sample preparation was carried using 1D$^2$ ligation kit LSK-308 and MinION flowcells R9.4. Each file assembly was sequenced in a separate flowcell and sequencing was carried out for 48 h.

## Data availability

All data are presented in the tables and figures of the paper. Sequencing data and raw gel images (Fig. 2c, d) can be found here: https://github.com/uwmisl/data-ncomms19-nanopore. Any additional data are available from the authors.

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

## Acknowledgements

This research was supported by a sponsored research agreement from Microsoft. The authors thank Oxford Nanopore Technologies for their technical assistance.

## Author contributions

R.L., Y.J.C., G.S., K.S., and L.C. conceived the assembly strategy and nanopore decoding scheme. R.L. and Y.J.C. designed the assembly sequences and carried out the experimental work. S.A., S.Y., K.M., M.Z.R., and K.S. developed the decoding scheme and decoded the files. R.L., Y.J.C., G.S., K.S., and L.C. wrote the paper. G.S, K.S., and L.C. oversaw the project.

## Additional information

**Competing Interests:** Y-J. C., S.D.A., and K.S. are Microsoft employees. The remaining authors declare no competing interests.

