## [Peer Review File · Nature Communications]

Reviewers' comments:

Reviewer #1 (Remarks to the Author):

Manuscript Overview

The manuscript outlines methodology to concatenate 150bp synthetic DNA sequences such that they are of sufficient length to enable more effective sequencing via Oxford Nanopore Technology (ONT) platforms. The usability of said methodology, which are presented as modified Gibson Assembly and Overlap-Extension PCR (OE-PCR), is demonstrated on the successful encoding and decoding of DNA sequences containing digital information of a totality of four files.

Review Summary:

The manuscript builds upon a concept of DNA concatenation previously published by the authors (Organick, L. et al. Random access in large-scale DNA data storage. Nat. Biotechnol. 2018). The data presented in the manuscript successfully demonstrates the functionality of the two concatenation approaches for standard sequencing on the ONT MinION. However, the manuscript as it is currently written appears to present these approaches as being novel and developed by the authors when they seem to incorporate and show similarity to many pre-existing techniques (Major Point 2). The OE-PCR approach as it is currently described appears similar to the pre-existing technique of the same name only differing in the content of the DNA sequences. There are many protocols, and modifications, for the joining of smaller DNA fragments and as such it is important that the manuscript clearly states how the presented approaches differ from and resemble existing techniques. It should be clear that the authors are applying pre-existing techniques (with clearly highlighted modifications if applicable) in a DNA storage context.

The manuscript requires addition discussion as to the need to concatenate small DNA sequences for ONT platforms (Major Point 1) and expanded methodology regarding RAM and encoding/decoding, as none is detailed, and primer design which is necessary for improved clarity (Major Point 3 and 4).

Major and minor concerns are outlined below.

Major Points:

1. The manuscript should include an expanded discussion on the need to concatenate short DNA sequences to enable sequencing on ONT platforms. As the authors state for scalable DNA storage systems it is necessary to use large oligo pool synthesis. In this reviewer's experience many companies now comfortably provide DNA sequences of length 200nt with this steadily increasing, not the 100-200nt that is stated on line 65. This length of 200nt is sufficient for sequencing on ONT platforms (line 103) as it is the minimum recommended size and therefore concatenation is not needed. Error can be mitigated by error correction coding and/or coverage. Concatenation may be useful to allow effective 'read until' selective sequencing (providing joining is sequential with appropriate RAM), however, the authors do not expand upon this or demonstrate this in the manuscript. In the current manuscript it is difficult to see how the Gibson-Assembly based approach would benefit from 'read until'.

2. It is important that the authors clearly highlight how this work differs from other modifications of Gibson Assembly and OE-PCR, and/or make it clear that the authors are applying pre-existing techniques in a DNA storage context. This is especially relevant for OE-PCR. With the current wording the authors appear to suggest that they have developed this assembly method, however, this technique is widely used in molecular biology to concatenate smaller DNA sequences. To this reviewer it does not appear to differ from the pre-existing technique of the same name, the incorporation of file ID is not especially novel as 3' and 5' modifications of varying nature are usually included. Also, the Gibson Assembly approach presented appears to include aspects of linker ssOligo modifications and NEBuilder HiFi DNA Assembly. Differences to existing techniques should be clearly highlighted, contrasted and cited.

3. Aspects of methodology for data encoding/decoding and RAM should be included as this is not fully detailed or cited. This reviewer has assumed that the strategies previously published by the authors have been adopted. However, if this is the case in addition to referencing previous publications a brief summary of the process should be included to assist those unfamiliar.

4. The manuscript is generally lacking in methodology details and clarity – especially relevant for the design of primers used for DNA assembly. It is hard to interpret how the primers for the 'sequential Gibson strategy' function based on the information provided and the schematic in Figure 2b. Is there controlled directionality in the way the DNA sequences encoding information (referred to as AD1 & AD2 in Figure 2b) are joined? The Methods section lacks sufficient detail such as, but not limited to, instrument details, DNA synthesis supplier and any bioinformatic tools used. In addition, aspects of the text are unclear. For instance, the grouping system related to lines 186-203 and Supplementary Fig 3 is confusing. Also, in some instances the number of files used in each approach (Gibson or OE-PCR) is ambiguous as it would appear that the incorrect number of files used is often stated.

5. Additional explanation is needed for the conditions at which NUPACK was used. The authors state at line 80 that primer pairs were tested at a 'starting concentration of 10nM at 25C', it is unclear how these conditions were selected. Why use 25C? Was a temperature gradient tested and 25C was simply the first temperature point? Are these specific to the Kapa HiFi enzyme mix buffer used? Furthermore, the testing of NUPACK with primer design appears selective. It would be useful to see how the authors strategy would scale/density and function with a range of conditions as nucleic acid thermodynamics vary with temperature and buffer usage.

Minor Points:

1. There are grammatical errors within the manuscript, particularly in figure legends.
2. Figure legend 1a-d and associated references in the text are in the incorrect order in relation to the labelled figures.
3. As the authors present this work as a Gibson Assembly based protocol the DG Gibson et., 2009 publication should be referenced.
4. In line 43 it is stated that the Illumina platform and associated workflow is 'poorly fitted to end-to-end automation'. It would be useful if the authors expanded on this statement as it is unclear why it is not amenable to automation. Many large-scale genomic centres have fully automated Illumina library preparation pipelines.
5. A MinION device contains 512 nanopore channels, in line 49 it is stated that these 'are connected to four biological nanopores', it should be clarified as to what are the four biological nanopores as this reviewer is uncertain.
6. Results, methodology and discussion is needed for the 'new consensus algorithm capable of handling higher error rates' (lines 83-84) implemented in the manuscript as this is only outlined.

Reviewer #2 (Remarks to the Author):

Lopez et al. present a proof of concept method for DNA data storage using long read sequencing with random access to selectively amplify select files. Their approach uses an improved decoding process that allows decoding at lower coverage. The authors cite the cost and latency of sequencing by synthesis as an advantage of portable nanopore sequencing. However, the latter still relies on short read synthesis, the primary bottleneck in current DNA storage solutions and sample preparation is more tedious with the additional assembly step. Further, due to current costs and read/write time, molecular data storage is most adaptable to low-touch data for long-term storage.

This study is a nice proof of principle but SBS is more amenable to current synthesis length limitations. Real time sequencing is a significant advantage over SBS but the authors do not discuss the scalability of their approach. The single Earth-Apollo file required 2 minION flowcells in order to generate a sufficient number of reads. The advantage of portability is outweighed by the limited scalability. I urge the authors to be more transparent about this limitation. Further ONT technology is indeed more portable than Illumina SBS but to say that it would “democratize [...] data storage technology outside of the research arena” is an exaggeration since the primary difference is in machine size and not accessibility to the equipment and reagents themselves.

The authors should read the manuscript closely as there were quite a lot of minor errors. The legend for Figure 1 does not entirely correspond to the figure itself.

It's not clear why the authors performed Gibson assembly as this method resulted in many fragments corresponding to smaller assemblies, requiring gel extraction. This approach is too tedious for scaling up an end-to-end DNA data storage. The authors present an alternative method using OE-PCR that allows better assembly and requires less tedious sequencing library preparation but the final assembly depends on the file synthesis. They mentioned other methods to assemble short reads. In fact, rolling circle amplification would be a more appropriate alternative but do not discuss the reason for their choice.

I suggest Supplementary Fig 3a be moved to the main text given that this is strategy is more scalable than the Gibson assembly approach, which the authors indeed discuss as inapplicable for a scalable storage workflow.

There is also no explanation or details about decoding using Illumina sequencing by synthesis. Given that the novelty of this study is a method that harnesses long read sequencing, it was not clear why they also performed decoding with SBS.

I would also like to remind the authors and editor of the Nature Communications policy for sharing all information, especially code, which the authors do not do.

“An inherent principle of publication is that others should be able to replicate and build upon the authors' published claims. A condition of publication in a Nature Research journal is that authors are required to make materials, data, code, and associated protocols promptly available to readers without undue qualifications. Any restrictions on the availability of materials or information must be disclosed to the editors at the time of submission. Any restrictions must also be disclosed in the submitted manuscript.”

Reviewers' comments:

Reviewer #1 (Remarks to the Author):

Manuscript Overview

The manuscript outlines methodology to concatenate 150bp synthetic DNA sequences such that they are of sufficient length to enable more effective sequencing via Oxford Nanopore Technology (ONT) platforms. The usability of said methodology, which are presented as modified Gibson Assembly and Overlap-Extension PCR (OE-PCR), is demonstrated on the successful encoding and decoding of DNA sequences containing digital information of a totality of four files.

Review Summary:

The manuscript builds upon a concept of DNA concatenation previously published by the authors (Organick, L. et al. Random access in large-scale DNA data storage. Nat. Biotechnol. 2018). The data presented in the manuscript successfully demonstrates the functionality of the two concatenation approaches for standard sequencing on the ONT MinION. However, the manuscript as it is currently written appears to present these approaches as being novel and developed by the authors when they seem to incorporate and show similarity to many pre-existing techniques (Major Point 2). The OE-PCR approach as it is currently described appears similar to the pre-existing technique of the same name only differing in the content of the DNA sequences. There are many protocols, and modifications, for the joining of smaller DNA fragments and as such it is important that the manuscript clearly states how the presented approaches differ from and resemble existing techniques. It should be clear that the authors are applying pre-existing techniques (with clearly highlighted modifications if applicable) in a DNA storage context.

The manuscript requires addition discussion as to the need to concatenate small DNA sequences for ONT platforms (Major Point 1) and expanded methodology regarding RAM and encoding/decoding, as none is detailed, and primer design which is necessary for improved clarity (Major Point 3 and 4).

Major and minor concerns are outlined below.

Major Points:

1. The manuscript should include an expanded discussion on the need to concatenate short DNA sequences to enable sequencing on ONT platforms. As the authors state for scalable DNA storage systems it is necessary to use large oligo pool synthesis. In this reviewers experience many companies now comfortably provide DNA sequences of length 200nt with this steadily increasing, not the 100-200nt that is stated on line 65. This length of 200nt is sufficient for sequencing on ONT platforms (line 103) as it is the minimum recommended size and therefore concatenation is not needed. Error can be mitigated by error correction coding and/or coverage. Concatenation may be useful to allow effective 'read until' selective sequencing (providing joining is sequential with appropriate RAM), however, the authors do not expand upon this or demonstrate this in the manuscript. In the current

manuscript it is difficult to see how the Gibson-Assembly based approach would benefit from ‘read until’.

Even though concatenation is not necessary, we found that it significantly increased the sequencing yield of the MinION platform. We clarified this point in the manuscript:

The overall yield and quality of a nanopore sequencing run is dependent on the molecular size of the DNA to be sequenced. DNA molecules translocate through the pore at a rate of 450 bases/sec while it can take between 2 to 4 seconds for a pore to capture and be occupied by the next DNA molecule. Therefore, short DNA molecules result in a higher number of unoccupied pores over time which increases the rate of electrolyte utilization above the membrane. This results in a faster loss in polarity and lower sequencing capacity and overall throughput. ONT estimates that the optimal DNA size to maximize sequencing yield is around 8 kilobases while the minimum size is 200 bases. Below 200 bases, event detection and basecalling is not possible. Therefore, a PCR-based random-access strategy for sequencing of the files encoded in 150 bp DNA oligonucleotides would have resulted in very low sequencing yield and limited decoding capabilities. Thus, we applied an assembly method 'Consecutive Gibson Assembly' to assemble large sequencing reads from short amplicons for the purpose of enabling higher nanopore sequencing throughput.

We found that increasing the input DNA size resulted in significant improvements in the sequencing throughput. To evaluate this improvement, we quantify sequencing throughput across 5 sequencing runs of equivalent quality. When comparing sequencing runs based on the input DNA fragment size, we found a modest reduction in the number of reads as the fragment size increased (Figure 5a). However, we found a 7-fold increase in sequencing throughput across the fragment size range we evaluated (between 600 and 4700 base pairs) (Figure 5b). This increase in sequencing yield was necessary to enable the successful decoding of a 1.5 MB file while sequencing using only two MinION flowcells.

2. It is important that the authors clearly highlight how this work differs from other modifications of Gibson Assembly and OE-PCR, and/or make it clear that the authors are applying pre-existing techniques in a DNA storage context. This is especially relevant for OE-PCR. With the current wording the authors appear to suggest that they have developed this assembly method, however, this technique is widely used in molecular biology to concatenate smaller DNA sequences. To this reviewer it does not appear to differ from the pre-existing technique of the same name, the incorporation of file ID is not especially novel as 3' and 5' modifications of varying nature are usually included. Also, the Gibson Assembly approach presented appears to include aspects of linker ssOligo modifications and NEBuilder HiFi DNA Assembly. Differences to existing techniques should be clearly highlighted, contrasted and cited.

For both concatenation strategies, we selected assembly methods that are used in molecular cloning and adopted them for the purpose of DNA storage. We revised the wording of the manuscript to make this clear and explained how we adapted existing workflow for a DNA storage application. We also added corresponding citations as suggested by the reviewer.

Thus, we applied an assembly method 'Consecutive Gibson Assembly' to assemble large sequencing reads from short amplicons for the purpose of enabling higher nanopore sequencing throughput (Figure 2a). Gibson Assembly is an isothermal and single-reaction method for assembly of multiple DNA sequences first developed by Daniel Gibson et. al. in 2009²². Since then, it has become a fundamental tool for the assembly of synthetic gene constructs in molecular cloning and synthetic biology due to its modularity and ease-of-use²³⁻²⁵. We re-purposed this tool for DNA storage to enable assembly and random access of digital files encoded in DNA pools containing multiple files in millions of oligonucleotides.

As an alternative to Gibson assembly, we explored the use of Overlap Extension PCR (OE-PCR) to assemble multiple oligonucleotides into larger DNA fragments (Figure 3). In the context of molecular cloning, OE-PCR is used as a simple approach to insert DNA fragments into plasmids or to join different gene fragments together^{27, 28}. To repurpose this approach for DNA storage, our strategy first splits the fixed-size DNA fragments of a file into multiple groups where each group has a unique forward ID and a unique back ID.

3. Aspects of methodology for data encoding/decoding and RAM should be included as this is not fully detailed or cited. This reviewer has assumed that the strategies previously published by the authors have been adopted. However, if this is the case in addition to referencing previous publications a brief summary of the process should be included to assist those unfamiliar.

We included additional citations on the adoption of random-access methodology for DNA storage. We briefly explained how we enable random access in the results section and we cited previous papers that implemented random access retrieval.

Our group and others have demonstrated random access by applying selective PCR amplification of a given file without having to read all the data stored in a particular DNA pool^{3, 8, 9, 14, 15}.

Our decoding process begins with a pool of synthetic DNA containing several files. Each file in the oligo pool consists of a set of 150-bases oligonucleotides with unique 20-nucleotide sequences at their 5' and 3' ends for PCR-based random-access retrieval (i.e. file ID) and a 110-nucleotide payload encoding the digital information. To amplify a specific file, PCR reactions are carried out with primer sets designed to amplify that file's ID, thereby enabling random access retrieval (Figure 2b).

4. The manuscript is generally lacking in methodology details and clarity – especially relevant for the design of primers used for DNA assembly. It is hard to interpret how the primers for the 'sequential Gibson strategy' function based on the information provided and the schematic in Figure 2b. Is there controlled directionality in the way the DNA sequences encoding information (referred to as AD1 & AD2 in Figure 2b) are joined? The Methods section lacks sufficient detail such as, but not limited to, instrument details, DNA synthesis supplier and any bioinformatic tools used. In addition, aspects of the text are unclear. For instance, the grouping system related to lines 186-203 and Supplementary Fig 3 is confusing. Also, in some instances the number of files used in each approach (Gibson or

OE-PCR) is ambiguous as it would appear that the incorrect number of files used is often stated.

We modified Figure 2 and corresponding caption to indicate the directionality of the Gibson Assembly:

First, a given file is PCR-amplified using primers specific to its address (AD_1 & AD_2) and containing overlapping overhang sequences (X_n). For a three-fragment assembly, three separate PCR amplification reactions are carried out, each containing primer pairs with different overhangs based on a sequential assembly design (X_1 & X_2^ , X_2 & X_3^* and X_3 & X_4^*). Upon amplification, products are purified and combined into by Gibson assembly. The assembly product can consist of any combination of oligonucleotides in a given file pool separated by ordered overhangs with a consistent directionality (e.g. X_1 - AD_1 -Payload₁- AD_2 - X_2 - AD_1 -Payload₂- AD_2 - X_3 - AD_1 -Payload₃- AD_2 - X_4).*

We clarified the mechanism of the Consecutive Gibson Assembly in the results section:

*Our decoding process begins with a pool of synthetic DNA containing several files. Each file in the oligo pool consists of a set of 150-bases oligonucleotides with unique 20-nucleotide sequences at their 5' and 3' ends for PCR-based random-access retrieval (i.e. file ID) and a 110-nucleotide payload encoding the digital information. To amplify a specific file, PCR reactions are carried out with primer sets designed to amplify a specific file's ID, thereby enabling random access retrieval (**Figure 2b**). To enable assembly, each file is amplified with multiple PCR reactions where each PCR reaction contains primer sets with the same file ID but with sequentially overlapping DNA overhangs such that the amplified fragments can be combined in a Gibson Assembly in a later step. Therefore, the number of PCR reactions depends on the chosen assembly size for a file.*

*Overhang sequence design was performed using a nucleic acid thermodynamic simulation software (NUPACK²⁶) to avoid primers with self-binding structures and cross-talk between orthogonal overhangs among other constraints (**Supplementary figure 2**). First, we generated a random set of 30-nucleotide DNA sequences with GC content between 40% and 60%, while avoiding 4-nucleotide repeats of G or C. Subsequently, we used NUPACK to estimate intramolecular secondary structure probability and selected those sequences with minimal secondary structure. This selection step was necessary to enable PCR reactions with comparable amplification efficiencies, which facilitated the generation of the assembly fragments. Next, we evaluated our overhang primer set for orthogonality by estimating binding probability across all possible primer pairs. Based on this cross-talk mapping, we then selected a subset of primers that exhibit minimum cross-talk among all pairs. We repeated this process to create overhangs for every new file to be assembled since each file contained different primer sequences. We expect that selecting an overhang sequence subset with minimal cross-talk can result in higher efficiency in the assembly process. We used this DNA sequence design method to generate overhang sequences for each file assembly.*

After PCR amplification of each file with its corresponding priming pairs, PCR products are combined, purified using magnetic beads and combined into a Gibson assembly reaction. During

the Gibson reaction, the exonuclease creates single-stranded 3' overhangs that enable the annealing of fragments that share complementarity at one end while the polymerase fills in gaps within each annealed fragment. Finally, a DNA ligase seals nicks in the assembled DNA. The product of the Gibson assembly was then amplified using primers corresponding to unique overhang sequences present at the ends of the assembly product. We implemented this approach to generate a 6-fragment assembly for each file, respectively. Using agarose gel electrophoresis, we found that the amplified Gibson Assembly product was the expected size (1,110 base pairs) with very small amounts of secondary products (Figure 2c).

We added additional details on the methods section regarding the PCR reaction reagents, protocol and instrumentation.

5. Additional explanation is needed for the conditions at which NUPACK was used. The authors state at line 80 that primer pairs were tested at a 'starting concentration of 10nM at 25C', it is unclear how these conditions were selected. Why use 25C? Was a temperature gradient tested and 25C was simply the first temperature point? Are these specific to the Kapa HiFi enzyme mix buffer used? Furthermore, the testing of NUPACK with primer design appears selective. It would be useful to see how the authors strategy would scale/density and function with a range of conditions as nucleic acid thermodynamics vary with temperature and buffer usage.

We elaborated on the sequence design methodology on the results and discussion sections.

Overhang sequence design was performed using a nucleic acid thermodynamic simulation software (NUPACK²⁶) to avoid primers with self-binding structures and cross-talk between orthogonal overhangs among other constraints (Supplementary figure 2). First, we generated a random set of 30-nucleotide DNA sequences with GC content between 40% and 60%, while avoiding 4-nucleotide repeats of G or C. Subsequently, we used NUPACK to estimate intramolecular secondary structure probability and selected those sequences with minimal secondary structure. This selection step was necessary to enable PCR reactions with comparable amplification efficiencies, which facilitated the generation of the assembly fragments. Next, we evaluated our overhang primer set for orthogonality by estimating binding probability across all primer pairs. Based on this cross-talk mapping, we then selected a subset of primers that exhibit minimum cross-talk among all pairs. We repeated this process to create overhangs for every new file to be assembled since each file contained different primer sequences. We expect that selecting an overhang sequence subset with minimal cross-talk can result in higher efficiency in the assembly process. We used this DNA sequence design method to generate overhang sequences for each file assembly.

Assembly sequence design. *Overhang sequence design was performed using a nucleic acid thermodynamic simulation software NUPACK v3.0.5. To evaluate primer self-binding, secondary structure analysis was performed using 'pairs' function from NUPACK at 65 °C to simulate the temperature of the annealing step in the PCR reaction. Resulting secondary structure predictions are ranked by probability of self-pairing and primers with low secondary structure are selected. To evaluate potential cross-talk between different overhangs in the*

assembly step, we evaluated cross-talk among all pairs of overhang primers using 'pairs' function from NUPACK at 25 °C, at 10nM of each overhang primer. Even though the Gibson Assembly reaction is carried out at 50°C, we set the temperature parameter at 25°C for a more stringent simulation. Next, pairs with an equilibrium binding of 1nM or above were sequentially eliminated until finding a subset of primers with minimal cross-talk.

Minor Points:

1. There are grammatical errors within the manuscript, particularly in figure legends.

The manuscript was carefully reviewed, and errors were corrected.

2. Figure legend 1a-d and associated references in the text are in the incorrect order in relation to the labelled figures.

The manuscript was carefully reviewed, and errors were corrected.

3. As the authors present this work as a Gibson Assembly based protocol the DG Gibson et., 2009 publication should be referenced.

A reference was added.

4. In line 43 it is stated that the Illumina platform and associated workflow is 'poorly fitted to end-to-end automation'. It would be useful if the authors expanded on this statement as it is unclear why it is not amenable to automation. Many large-scale genomic centers have fully automated Illumina library preparation pipelines.

We revised the text to remove this description.

5. A MinION device contains 512 nanopore channels, in line 49 it is stated that these 'are connected to four biological nanopores', it should be clarified as to what are the four biological nanopores as this reviewer is uncertain.

We clarified this point:

Nanopore sequencing, as commercialized by Oxford Nanopore Technologies (ONT), offers a sequencing alternative that is portable, inexpensive and automation-friendly, resulting in a better alternative for a real-time "read head" of a molecular storage system¹⁶⁻²⁰. Specifically, ONT MinION is a four-inch long USB-powered device containing an array of 512 sensors, each connected to four biological nanopores, capable of producing up to 15 gigabases of sequencing output per flowcell²¹. Each nanopore is built into an electrically resistant artificial membrane. During sequencing, a single strand of DNA passes through the pore resulting in a change in the current across the membrane. This electrical signal is processed in real time to determine the sequence identity of the DNA strand. In the context of DNA storage, real-time sequencing enables the ability to sequence until sufficient coverage has been acquired for successful decoding without having to wait for an entire sequencing run to be completed.

For additional explanation, we included an additional reference describing in more detail the design and function of the MinION device:

Lu, H.; Giordano, F.; Ning, Z. Oxford Nanopore MinION Sequencing and Genome Assembly. *Genomics, proteomics & bioinformatics* 2016, 14, 265–279.

6. Results, methodology and discussion is needed for the ‘new consensus algorithm capable of handling higher error rates’ (lines 83-84) implemented in the manuscript as this is only outlined.

The changes in the consensus algorithm for handling nanopore reads are described in the results section.

In our previous algorithm²⁹, the consensus sequence is recovered by a process where pointers for payload sequences are maintained and moved from left to right, and at every stage of the process the next symbol of the sequence is estimated via a plurality vote. For payload sequences that agree with plurality, the pointer is moved to the right by 1. But for the sequences that do not agree with plurality, the algorithm classifies whether the reason for the disagreement is a single deletion, an insertion, or a substitution. This is done by looking at the context around the symbol under consideration. Once this is estimated, the pointers are then moved to the right accordingly.

The key difference between our new algorithm and our previous implementation is that in cases when disagreements cannot be classified, we do not drop respective payload sequences from further consideration. Instead, we label such payload sequences as being out of sync and attempt to bring them back at later stages. Specifically, every sequence that is out of sync is ignored for several next steps of the algorithm. However, after those steps, we perform search for a match between the last few bases of the partially constructed consensus sequence and appropriately located short substrings of the payload sequence. If we discover a match, we move the payload sequence pointer to the corresponding location and drop the out of sync label. This allows the payload sequence pointer to circumvent small groups of adjacent incorrect bases, a feature that was not present in the earlier algorithm. This modification to the consensus algorithm allows us to successfully decode from notably lower coverages because more information from the sequencing reads is used in the process.

Reviewer #2 (Remarks to the Author):

Lopez et al. present a proof of concept method for DNA data storage using long read sequencing with random access to selectively amplify select files. Their approach uses an improved decoding process that allows decoding at lower coverage. The authors cite the cost and latency of sequencing by synthesis as an advantage of portable nanopore sequencing. However, the latter still relies on short read synthesis, the primary bottleneck in current DNA storage solutions and sample preparation is more tedious with the

additional assembly step. Further, due to current costs and read/write time, molecular data storage is most adaptable to low-touch data for long-term storage.

This study is a nice proof of principle but SBS is more amenable to current synthesis length limitations. Real time sequencing is a significant advantage over SBS but the authors do not discuss the scalability of their approach. The single Earth-Apollo file required 2 minION flowcells in order to generate a sufficient number of reads. The advantage of portability is outweighed by the limited scalability. I urge the authors to be more transparent about this limitation. Further ONT technology is indeed more portable than Illumina SBS but to say that it would “democratize [...] data storage technology outside of the research arena” is an exaggeration since the primary difference is in machine size and not accessibility to the equipment and reagents themselves.

We revised the following paragraph in the discussion section to clarify the promise and limitations of nanopore sequencing for DNA storage:

The future of DNA data storage will depend on technologies that enable cost-efficient and fast reading and writing synthetic DNA. While sequencing by synthesis remains widely used for its low error rate and reliability, nanopore sequencing is being rapidly adopted in applications that require long DNA sequencing reads³¹, high degree of portability¹⁴, or real-time sequencing¹⁵. High error rates and limited sequencing throughput remain significant limitations for this technology to be applicable in a scalable DNA storage workflow. Nevertheless, the commercial implementation of this technology is still in its infancy and future technological improvements can enable a nanopore-based read-head for DNA storage to become cost-effective and widely adopted. Specifically, recent advances in base-calling algorithms³², improved biological nanopores³³, and the emergence of solid-state nanopores³⁴ promise to further improve the cost and scale of nanopore sequencing.

The authors should read the manuscript closely as there were quite a lot of minor errors. The legend for Figure 1 does not entirely correspond to the figure itself.

The manuscript was carefully reviewed, and errors were corrected, including that in Figure 1’s legend.

It’s not clear why the authors performed Gibson assembly as this method resulted in many fragments corresponding to smaller assemblies, requiring gel extraction. This approach is too tedious for scaling up an end-to-end DNA data storage. The authors present an alternative method using OE-PCR that allows better assembly and requires less tedious sequencing library preparation but the final assembly depends on the file synthesis. They mentioned other methods to assemble short reads. In fact, rolling circle amplification would be a more appropriate alternative but do not discuss the reason for their choice.

We edited our manuscript in the discussion section to include the limitations of our assembly strategy and our experience with rolling circle amplification:

We found that assembling large DNA fragments from short DNA oligonucleotides was crucial for increasing nanopore sequencing throughput and decoding large digital files stored in DNA. However, both assembly methods required additional sample preparation steps compared to our previous work⁷ where random access and sample preparation for SBS was enabled by a single PCR reaction from the original pool. As an alternative, we experimented with rolling circle amplification (RCA) since it had been previously described as an easier alternative for building assemblies for nanopore sequencing³⁰. However, we found a reduction in sequencing throughput when using the product of an RCA reaction compared to the other assembly methods. A potential explanation for reduction in throughput is that the branched nature of the RCA product hinders the translocation of the DNA through the nanopore resulting in fewer sequencing reads. In addition to these alternatives, there are many other DNA assembly methods that could be adapted into a nanopore sequencing sample preparation workflow and may enable higher sequencing throughput with more efficient sample preparation.

I suggest Supplementary Fig 3a be moved to the main text given that this is strategy is more scalable than the Gibson assembly approach, which the authors indeed discuss as inapplicable for a scalable storage workflow.

We moved Supplementary Fig 3. to the main text.

There is also no explanation or details about decoding using Illumina sequencing by synthesis. Given that the novelty of this study is a method that harnesses long read sequencing, it was not clear why they also performed decoding with SBS.

Every file included in this publication was previously sequenced and decoded using SBS without the need for assembly. Given the limitations in sequencing length for SBS, we did not sequence the assembly products with SBS but used our previous results as comparison. We revised the text to clarify this point:

In a previous publication from our group, we had demonstrated successful decoding of all these files without assembly using an Illumina sequencing platform⁷.

I would also like to remind the authors and editor of the Nature Communications policy for sharing all information, especially code, which the authors do not do. “An inherent principle of publication is that others should be able to replicate and build upon the authors' published claims. A condition of publication in a Nature Research journal is that authors are required to make materials, data, code, and associated protocols promptly available to readers without undue qualifications. Any restrictions on the availability of materials or information must be disclosed to the editors at the time of submission. Any restrictions must also be disclosed in the submitted manuscript.”

The encoding algorithm is now described in the methods section of the manuscript. This algorithm is identical to the one used in our previous publication and a reference was included.

Encoding files into DNA. Our encoding process starts by randomizing data to reduce chances of secondary structures, primer–payload non-specific binding, and improved properties during

decoding. It then breaks the data into fixed-size payloads, adds addressing information, and applies outer coding, which adds redundant sequences using a Reed–Solomon code to increase robustness to missing sequences and errors. The level of redundancy is determined by expected errors in sequencing and synthesis, as well as DNA degradation. Next, it applies inner coding, which converts the bits to DNA sequences. The resulting set of sequences is surrounded by a primer pair chosen from the library based on low level of overlap with payloads. A more detailed description of the encoding algorithm can be found in our previous publication⁷. The resulting collection of sequences is synthesized by Twist Bioscience. Upon arrival, DNA is rehydrated the pool in $1 \times TE$ buffer and stored at $-20\text{ }^{\circ}\text{C}$.

The decoding algorithm is now described in the results section of the manuscript and an associated patent with a more detailed description was cited. We modified the manuscript to describe the modifications in the algorithm for decoding nanopore sequencing reads.

*We performed an equivalent sequencing analysis for the other three files (**Supplementary Fig. 3, 4 and 5**) and then we attempted decoding each file using the $1D^2$ reads as inputs to a decoder algorithm. Since each read contains multiple payload sequences, we performed multiple sequential alignments of the front and back file ID in each read and then selected the sequence in between, which should correspond to the payload sequences. The decoding process starts by clustering these payload sequences based on similarity and finding a consensus between the sequences in each cluster to reconstruct the original sequences. Once a consensus has been reached, these sequences are decoded back to digital data to retrieve the original file. We used a decoder algorithm from our previous work but modified the consensus sequence algorithm to better utilize nanopore sequencing data.*

In our previous algorithm²⁹, the consensus sequence is recovered by a process where pointers for payload sequences are maintained and moved from left to right, and at every stage of the process the next symbol of the sequence is estimated via a plurality vote. For payload sequences that agree with plurality, the pointer is moved to the right by 1. But for the sequences that do not agree with plurality, the algorithm classifies whether the reason for the disagreement is a single deletion, an insertion, or a substitution. This is done by looking at the context around the symbol under consideration. Once this is estimated, the pointers are then moved to the right accordingly.

The key difference between our new algorithm and our previous implementation is that in cases when disagreements cannot be classified, we do not drop respective payload sequences from further consideration. Instead, we label such payload sequences as being out of sync and attempt to bring them back at later stages. Specifically, every sequence that is out of sync is ignored for several next steps of the algorithm. However, after those steps, we perform search for a match between the last few bases of the partially constructed consensus sequence and appropriately located short substrings of the payload sequence. If we discover a match, we move the payload sequence pointer to the corresponding location and drop the out of sync label. This allows the payload sequence pointer to circumvent small groups of adjacent incorrect bases, a feature that was not present in the earlier algorithm. This modification to the consensus algorithm allows us to successfully decode from notably lower coverages because more information from the sequencing reads is used in the process.

REVIEWERS' COMMENTS:

Reviewer #1 (Remarks to the Author):

General comment following revised manuscript and authors rebuttal:

With the exception of two points outlined below the authors have appropriately addressed this reviewers major and minor points of concern. Remaining points of concern are simply editorial errors and do not refer to the scientific content.

Regarding Major Point 2:

The majority of the manuscript has been revised to emphasise that pre-existing concatenation methodology has been applied to concatenate 150bp synthetic DNA sequences such that they are of sufficient length to enable more effective sequencing via Oxford Nanopore Technology (ONT) platforms. However, in the abstract it is still stated that a 'novel assembly strategy' has been developed by the authors which to this reviewer is not a true claim.

Regarding Minor Point 1 and 2:

Previous errors have been corrected, however, there are errors within the text and figure legends of Figures 2 and 3. The text relating to Figure 2 appears out of synch with the figure order. On line 166 the authors refer to Figure 2e – this figure is omitted from the manuscript. Similarly, the text relating to Figure 3 may also be out of synch.

Reviewer: Emily E Hesketh, Wellcome Sanger Institute

Reviewer #2 (Remarks to the Author):

All concerns by myself and the other reviewer appear to have been addressed, importantly, the detailed molecular and computational methods and transparency about the scalability of the assembly approach. Overall, this study shows modest improvements over their previous publication (Organick et al. 2018) and the OE-PCR/random access approach by Yazdi et al. 2015 & 2017. The approach primarily increases the scale of random access through assembly of short amplicons and an improved consensus algorithm to handle the high error rates of nanopore sequencing. The assembly technique necessary for the use of nanopore sequencing is limiting but demonstrates a proof of concept for long read technology which will be necessary for read-until random access of DNA storage.